# Boosting Unsupervised Contrastive Learning Using Diffusion-Based Data Augmentation From Scratch

## Abstract

Unsupervised Contrastive learning has gained prominence in fields such as vision, natural language processing, and biology, leveraging predefined positive and negative samples for representation learning. Data augmentation, categorized into hand-designed and model-based methods, has been identified as a crucial component for enhancing contrastive learning. However, hand-designed methods require human expertise in domain-specific data while sometimes distorting the meaning of the data. In contrast, model-based methods, such as generative models, usually require supervision or large-scale external data. To address the problems presented above, this paper proposes DiffAug, a novel unsupervised contrastive learning technique with diffusion mode-based positive data generation. DiffAug consists of a semantic encoder and a conditional diffusion model; the conditional diffusion model generates new positive samples conditioned on the semantic encoding to serve the training of unsupervised contrast learning. With the help of iterative training of the semantic encoder and diffusion model, DiffAug improves the representation ability in an uninterrupted and unsupervised manner. Experimental evaluations show that DiffAug outperforms hand-designed augmentation and classical representation learning methods in classification and clustering tasks on visual and biological datasets, highlighting its potential for generalizing unsupervised learning techniques. The code for review is released at https://anonymous.4open.science/r/diffaug_review-804E.

## 1 Introduction

Contrastive learning, as shown by many studies (He et al., 2020; Chen et al., 2020; Cui et al., 2021; Wang & Qi, 2022; Assran et al., 2022; Zang et al., 2023), has become important in areas like vision (He et al., 2021; Zang et al., 2022b), natural language processing (Rethmeier & Augenstein, 2023), and biology (Yu et al., 2023; Krishnan et al., 2022). It learns representations using predefined positive and negative samples. Many studies (Tian et al., 2020; Zhang & Ma, 2022; Peng et al., 2022; Zhang et al., 2023b) have found that data augmentation helps contrastive learning by making it more robust and preventing model problems.

Data augmentation falls into two main types: *hand-designed methods* and *model-based methods* (Xu et al., 2023). Hand-designed methods require humans to understand the meaning of the data and then change the input features while maintaining or extending that meaning. In visual tasks, methods such as color change (Yan et al., 2022), random cropping (Cubuk et al., 2020), and rotation (Maharana et al., 2022) are used to aid in contrastive learning. However, the problem is that the above techniques must be more data-specific. For some data (genes or proteins or others), it isn't easy to visualize the data due to the complexity of its meaning. Consequently, it isn't easy to design a good augmentation strategy. Semantics-independent augmentation methods such as adding noise (Huang et al., 2022) and random hiding (Theodoris et al., 2023) are used, but only sometimes with significant results. Another problem with hand-designed methods is that they do not smoothly change the semantics of the data, e.g., a slight change in the magnitude of the magnification in a random cropping of an image may imply the risk of a semantic mutation (in Fig. 1). As a result, many positive and negative samples are needed to distribute these risks to obtain a stable representation. At the same time, it is

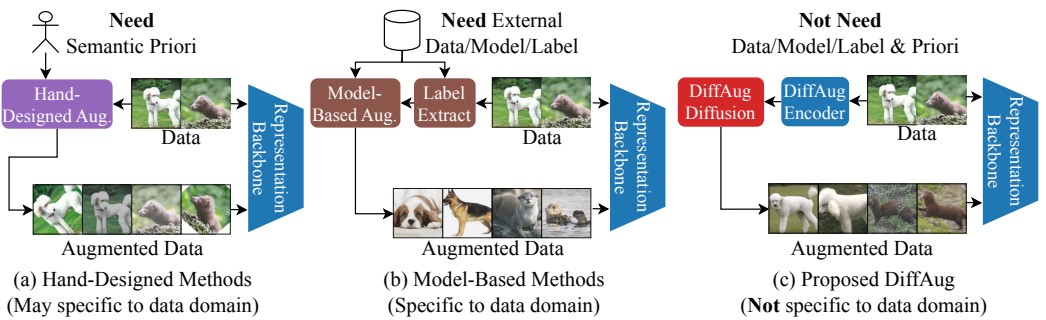

Figure 1: **Comparison of DiffAug with other methods.** (a) Hand-designed augmentation methods are based on human a priori design strategies that produce new data that differ in input features and are semantically similar. (b) Model-based augmentation methods are mainly based on generative models that produce new data with the same labels with the help of a large amount of data, labels, or pre-trained models. Such methods tend to be specific to the data domain. (c) DiffAug attempts to reduce the dependence on external data and prior knowledge through iterative training with encoders and diffusion. Expanding the application areas of unsupervised contrastive learning.

also challenging to train contrastive learning models with fewer samples for certain domains where data acquisition is costly, such as biology.

Given the challenges mentioned earlier, model-based methods (generative models) based on deep learning are used to create better data. In the vision domain, techniques using VAE (Kingma & Welling, 2014), GAN (Goodfellow et al., 2014), and diffusion models (Ho et al., 2020; Nichol & Dhariwal, 2021; Saharia et al., 2022; Nichol et al., 2022; Ramesh et al., 2022) have been developed to improve model training. For supervised learning, several studies have received attention. Du et al. (2023) proposed the DREAM-OOD framework, which uses diffusion models to generate photo-realistic outliers from in-distribution data for improved OOD detection. Zhang et al. (2023a) developed the Guided Imagination Framework (GIF) using generative models like DALL-E2 and Stable Diffusion for dataset expansion, enhancing accuracy in both natural and medical image datasets. However, there are concerns about these methods, especially about their diversity and how well they generalize. The detailed related works are in Appendix.A. Moreover, most of these generative models are trained with supervision or need much external data. This makes them less suitable for areas like gene and protein data (in Fig. 1). This leads to an important question: **Is it possible to design a data augmentation framework to enhance unsupervised contrastive learning in different domains without requiring expert knowledge or additional data?**

We introduce DiffAug, a novel unsupervised contrastive learning technique with diffusion mode-based positive data generation to address the posed question. Explicitly designed for unsupervised contrastive learning, DiffAug eliminates the need for training labels. Instead, we employ a semantic estimator to gauge the semantics of the input data, subsequently guiding the augmentation process. At its core, DiffAug operates through two synergistic modules: a semantic encoder and a diffusion generator. Utilizing a soft contrastive loss, the semantic encoder crafts latent representations that act as guiding vectors for the diffusion generator. This generator then methodically produces augmented data in the input space, ensuring varying levels of semantic consistency based on the guiding vectors and specific adjustable hyperparameters.

In our experiments, we thoroughly evaluated our method on both visual and biological datasets. Our findings indicate that the proposed method can produce sensible data augmentations, subsequently enhancing the performance of unsupervised contrastive learning that utilizes these augmentations. Notably, DiffAug performs superior classification and clustering tasks compared to all benchmark methods. The primary contributions of this paper are: (a) We introduce DiffAug, a novel unsupervised contrastive learning technique with diffusion mode-based positive data generation. DiffAug's data augmentation replaces traditional domain-specific hand-designed data augmentation strategy. (b) DiffAug operates independently of external data or manually designed rules. Its versatility allows for integration with various models, encompassing domains like vision or biology studies. (b) The experimental results show the efficacy of DiffAug in enhancing the performance of contrastive

learning in different tasks. This suggests that DiffAug can generate positive sample data unsupervised, which in turn promotes the development of unsupervised learning techniques.

## 2 How to Build Unsupervised Conditional Generative Models

In the context of unsupervised data augmentation, the training dataset providing potential semantic categories is denoted as $\mathcal{D}_t = \{\mathbf{x}_i\}_{i=1}^N$, where $N$ is the size of the training set. To boost the training efficiency of unsupervised contrastive learning with positive samples generated by the diffusion model, a novel framework called DiffAug is proposed.

### 2.1 Preliminaries of Contrastive Learning and Soft Contrastive Learning

**Contrastive Learning.** Contrastive learning learns visual representation via enforcing the similarity of the positive pairs and enlarging distance of negative pairs. Formally, loss is defined as,

$$\mathcal{L}_{\mathrm{cl}} = -\log \mathcal{Q}\left(\mathbf{z}_i, \mathbf{z}_i^+\right) + \log\left[\mathcal{Q}\left(\mathbf{z}_i, \mathbf{z}_i^+\right) + \sum_{\mathbf{z}_i^- \in V^-} \mathcal{Q}\left(\mathbf{z}_i, \mathbf{z}_i^-\right)\right] \tag{1}$$

where $\mathbf{z}_i$ is the low dimensional embedding $\mathbf{z}_i = \mathrm{Enc}^{\mathrm{cl}}(\mathbf{x}_i)$, $Q\left(\mathbf{z}_i, \mathbf{z}_i^+\right)$ indicates the similarity between positive pairs while $Q\left(\mathbf{z}_i, \mathbf{z}_i^-\right)$ is the similarity between negative pairs. For the traditional scheme, in the computer vision domain, data augmentation methods such as random cropping (Cubuk et al., 2020) or data mixup (Zhang et al., 2017) are used to generate new positive data. The negative samples $v_i^-$ are sampled from negative distribution $V^-$.

**Soft Contrastive Learning.** To address the performance degradation due to view noise in contrastive learning and to accomplish unsupervised learning on smaller scale datasets, Zang et al. (2023) designed soft contrastive learning, which smoothes sharp positive and negative sample pair labels by evaluating the credibility of the sample pairs. Consider the loss form for multiple positive samples and multiple negative samples as,

$$\mathcal{L}_{\mathrm{scl}}(\mathbf{y}_c, \mathbf{y}_j, \mathbf{z}_c, \mathbf{z}_j) = -\sum_{j=1}^{\mathcal{B}} \{\underbrace{\mathcal{P}(\mathbf{y}_c, \mathbf{y}_j)}_{\text{soft positive aim}} \log\left(\mathcal{Q}(\mathbf{z}_c, \mathbf{z}_j)\right) + \underbrace{(1 - \mathcal{P}(\mathbf{y}_c, \mathbf{y}_j))}_{\text{soft negative aim}} \log\left(1 - \mathcal{Q}(\mathbf{z}_c, \mathbf{z}_j)\right)\},$$

$$\mathcal{P}(\mathbf{a}, \mathbf{b}) = \left(1 + \mathcal{H}_{ij}\left(e^\beta - 1\right)\right) \mathcal{Q}(\mathbf{a}, \mathbf{b}), \tag{2}$$

where the $\mathbf{y}_i, \mathbf{z}_i$ are the high dimensional embedding and low dimensional embedding $\mathbf{y_i}, \mathbf{z_i} = \mathrm{Enc}(\mathbf{x_i})$. The $\mathcal{P}(\mathbf{a}, \mathbf{b})$ is soft learning weight and calculated by the positive/negative pair indicator $\mathcal{H}_{cj}$. The hyper-parameter $\beta \in [0, 1]$ introduces prior knowledge of data augmentation relationship $\mathcal{H}_{cj}$ into the model training. Details of contrastive and soft contrastive learning are in Appendix B.

### 2.2 DiffAug Design Details and Training Strategies

**DiffAug Framework.** DiffAug accomplishes the tasks of *positive sample generation* and *data representation* by iterating the two modules over each other (in Fig. 2). DiffAug consists of two main modules, a semantic encoder $\mathrm{Enc}(\cdot|\theta)$ and a diffusion generator $\mathrm{Gen}(\cdot|\phi)$, where $\theta$ and $\phi$ are model parameters. The $\mathrm{Enc}(\cdot|\theta)$ maps the input data $\mathbf{x}_i$ to the discriminative latent space $\mathbf{v}_i$, and the generator $\mathrm{Gen}(\cdot|\phi)$ generates new data with a semantic vector $\mathbf{v}_i$. Similar to the Expectation maximization algorithm (Gupta et al., 2011), the semantic encoder $\mathrm{Enc}(\cdot|\theta)$ and the diffusion generator $\mathrm{Gen}(\cdot|\phi)$ are trained in turn by two different loss functions (see Fig. 2(a) and Fig. 2(b)).

**Semanticity Modeling (E-Step).** In the semanticity modeling step, given a central data $\mathbf{x}_c$, we generate a background set $\mathcal{B}_c$,

$$\mathcal{B}_c = \{\mathbf{x}_1, \cdots, \mathbf{x}_j, \cdots, \mathbf{x}_{\mathcal{N}_b}\}, \begin{cases} \mathbf{x}_j \sim \mathcal{D}_t & \text{if } \mathcal{H}_{cj} = 0 \\ \mathbf{x}_j \sim A_{\mathrm{ug}}(\mathbf{x}_c) & \text{if } \mathcal{H}_{cj} = 1 \end{cases} \tag{3}$$

where $N_b$ is the number of background data points. The $\mathcal{H}_{cj} = 0$ indicates $\mathbf{x}_j$ is sampled from the dataset $\mathcal{D}_t$, and $\mathbf{x}_c$ and $\mathbf{x}_j$ are negative pair. Meanwhile, $\mathcal{H}_{cj} = 1$ indicates $\mathbf{x}_c$ and $\mathbf{x}_j$ are positive

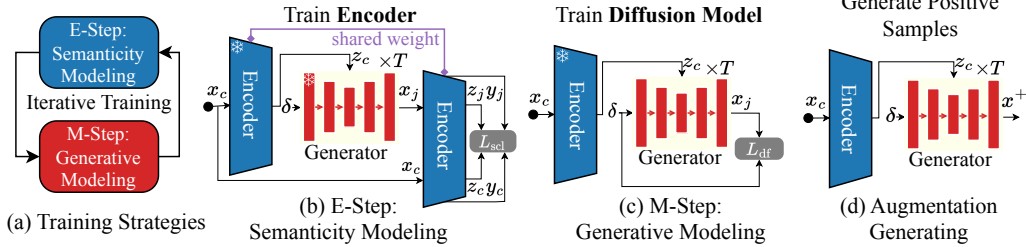

Figure 2: **The DiffAug framework and training strategy.** DiffAug includes a semantic encoder $\text{Enc}(\cdot|\theta)$ and a diffusion generator $\text{Gen}(\cdot|\phi)$. (a) shows how $\text{Enc}(\cdot|\theta)$ and $\text{Gen}(\cdot|\phi)$ are interative trained. (b) and (c) show how to calculate the loss functions. (d) shows how to generate new augmentation data with the trained model. The snowflake (❄) indicates that the module is frozen.

pair and $\mathbf{x}_j$ is sampled from data augmentation. For details, new positive data are generated by the diffusion model according to DDMP (Ho et al., 2020),

$$\mathbf{x}_j = \text{Gen}(\delta, \mathbf{z}_c|\phi^*), \mathbf{y}_c, \mathbf{z}_c = \text{Enc}(\mathbf{x_c}|\theta^*), \tag{4}$$

where $\text{Gen}(\delta, \mathbf{z}_c|\phi^*)$ is the generation process of the diffusion model, and the generating details are in Eq. (7). The $\delta \sim \mathcal{N}(0, \mathbf{1})$ is the random initialized data, and $\mathbf{z}_c$ is a conditional vector. The $*$ in $\phi^*$ and $\theta^*$ means the parameter is frozen. To avoid unstable positive samples from untrained generative models, training starts exclusively with traditional data augmentation tools, and then, the data generated by DiffAug is replaced with data generated by DiffAug, with a replacement probability of the hyperparameter $\lambda$, an oversized $\lambda$ introduce toxicity, which we will discuss in Sec. 3.5. We update the parameter of the semantic encoder with the soft contrastive learning loss,

$$\theta = \theta - \eta \sum_{\mathbf{x}_j \in \mathcal{B}_c} \{\mathcal{L}_{\text{scl}}(\mathbf{y}_c, \mathbf{z}_c, \mathbf{y}_j, \mathbf{z}_j)\}, \text{ where } \mathbf{y}_j, \mathbf{z}_j = \text{Enc}(\mathbf{x}_j|\theta), \tag{5}$$

where the $\eta$ is the learning rate, and the $\mathcal{L}_{\text{scl}}(\mathbf{y}_c, \mathbf{z}_c, \mathbf{y}_j, \mathbf{z}_j)$ is in Eq. (2).

**Generative Modeling (M-Step).** In the generative modeling step, the conditional diffusion generator $\text{Gen}(\cdot|\phi)$ is trained by the vanilla diffusion loss $\mathcal{L}_{\text{df}}(\mathbf{x}_c, \mathbf{z}_c|\phi)$ (Ho et al., 2020),

$$\phi = \phi - \eta \sum_{t=1}^{T} \left\{ \left\| \delta - g_\phi \left( \sqrt{\bar{\alpha}_t} \widetilde{\mathbf{x}}_c^t + \sqrt{1 - \bar{\alpha}_t}, t, \mathbf{z}_c \right) \right\|_2^2 \right\}, \tag{6}$$

where the conditional vector $\mathbf{z}_c$ is generated from the semantic encoder in Eq.(4). The $g_\phi(\cdot)$ is the conditional diffusion neural network. The $\alpha_t$ is the noise parameter in the diffusion process, and $\bar{\alpha}_t = 1 - \alpha_t$. The $\widetilde{\mathbf{x}}_c^t$ is the intermediate data in the diffusion process, and the $\widetilde{\mathbf{x}}_c^0 = \mathbf{x}_c$. $T$ is the time step of the generation process. When $g_\phi(\cdot)$ is trained, the detailed generating process is,

$$\text{Gen}(\delta, \mathbf{z}_c|\phi^*) = \left\{ \widetilde{\mathbf{x}}^0 \mid \widetilde{\mathbf{x}}^{t-1} = \frac{1}{\sqrt{\alpha_t}} \left( \widetilde{\mathbf{x}}^t - \frac{1 - \alpha_t}{\sqrt{1 - \bar{\alpha}_t}} g_\phi(\widetilde{\mathbf{x}}^t, t, \mathbf{z}_c^*) \right) + \sigma_t \mathcal{N}(0, 1), t \in \{T, \cdots, 1\} \right\}, \tag{7}$$

The $g_\phi(\cdot)$ is a neural network approximator intended to predict $\delta$ with $\widetilde{\mathbf{x}}$ and the condition vector $\mathbf{z}_c^*$.

**Augmentation Generation** Given the trained semantic encoder $\text{Enc}(\cdot|\theta)$ and diffusion generator $D(\cdot)$, and DiffAug generate new augmented data $\mathbf{x}_i^+$ from any input data $\mathbf{x}_i$.

$$\mathbf{x}_i^+ = \text{Gen}(\delta|\mathbf{z}_i), \ \mathbf{y}_i, \mathbf{z}_i = \text{Enc}(\mathbf{x}_i). \tag{8}$$

Meanwhile, DiffAug's semantic encoder can be seen as a feature extractor. It is considered to have good discriminative performance because it is trained simultaneously as the diffusion generator.

## 3 RESULTS

We completed experiments on both the vision dataset and the biological dataset. We want to prove that Diffaug can work effectively and bring enhancement in different domains.

---

**Algorithm 1** The DiffAug Training Algorithm:

---

**Input**: Data: $\mathcal{D}_t = \{\mathbf{x}_c\}_{i=1}^{N}$, Learning rate: $\eta$, E or M State: S, Batch size: $B$, Network parameters: $\theta, \phi$,

**Output**: Updateed Parameters: $\theta, \phi$.

1: **while** $b = 0; b < [|\mathcal{X}|/B]; b$++ **do**
2:     $\mathbf{x}_c \sim \mathcal{D}_t$;                         # Sample the centering data
3:     $\mathbf{y}_c, \mathbf{z}_c \leftarrow \text{Enc}(\mathbf{x}_c|\theta)$;             # Generate frozen condition vector
4:     **if** S==M-step **then**
5:        $L_1 \leftarrow L_{\text{df}}(\mathbf{x}_c, \text{SG}(\mathbf{z}_c)|\phi)$ by Eq. (6); $\phi \leftarrow \phi - \eta \frac{\partial \mathcal{L}_1}{\partial \phi}$,    # Calculate diffusion loss
6:     **else**
7:        $\mathcal{B}_c = \{\mathbf{x}_1, \cdots, \mathbf{x}_{\mathcal{B}} | \mathbf{x}_j \sim \mathcal{D}_t \text{ if } \mathcal{H}_{ij} = 0; \mathbf{x}_j \sim \text{Aug}(\mathbf{x}_c) \text{ else }\}$; # Generate/sample data
8:        $\mathcal{Y} = \{\mathbf{y}_1, \cdots, \mathbf{z}_j, \cdots, \mathbf{y}_{\mathcal{B}}\}$, $\mathcal{Z} = \{\mathbf{z}_1, \cdots, \mathbf{z}_j, \cdots, \mathbf{z}_{\mathcal{B}}\}$, $\mathbf{y}_j, \mathbf{z}_j = \text{Enc}(\mathbf{x}_j|\theta)$
9:        $L_2 \leftarrow L_{\text{scl}}(\mathcal{Y}, \mathcal{Z})$ by Eq. (2); $\theta \leftarrow \theta - \eta \frac{\partial \mathcal{L}_2}{\partial \theta}$          # Calculate scl loss
10:    **end if**
11: **end while**

---

Table 1: **Comparison of Linear-test and KMeans clustering performance on computer vision dataset.** The contrastive learning methods, including SimCLR (Chen et al., 2020), MOCO v2 (He et al., 2020), BYOL (Grill et al., 2020), SimSiam (Chen & He, 2021), and DLME (Zang et al., 2022b) are chosen for comparison. The SimC.+Mix. and MoCo.+Mix. are SimCLR and Mo-CoV2 with Mixup data augmentation which processed by Zhang et al. (2022). The SimC.+Dif. and MoCo.+Mix. are SimCLR and MoCoV2 with DiffAug data augmentation. Improvements over the best baseline are shown in parentheses. The 'AVE' represents the average of the performance on all datasets. The best results are marked in **bold**. Performance gains of more than 1.0 are underlined.

| | | Linear-test Performance | | | Clustering Performance | | | | |
|---|---|---|---|---|---|---|---|---|---|
| | CF10 | CF100 | STL10 | TINet | CF10 | CF100 | STL10 | TINet | AVE |
| SimCLR | 89.8 | 57.4 | 86.9 | 38.4 | 78.2 | 38.6 | 77.2 | 12.9 | 58.6 |
| MoCoV2 | 90.1 | 64.2 | 85.6 | 39.4 | 79.9 | 36.4 | 75.4 | 14.4 | 59.3 |
| BYOL | 91.0 | 62.7 | 88.7 | 43.8 | 82.6 | 45.6 | 78.5 | 18.8 | 62.6 |
| SimSiam | 90.6 | 63.5 | 84.8 | 44.9 | 79.5 | 42.1 | 82.8 | 18.9 | 62.5 |
| DLME | 91.3 | 66.1 | 90.1 | 44.9 | 83.1 | 44.1 | 88.3 | 18.2 | 64.9 |
| SimC.+Mix. | 90.9 | 62.9 | 89.6 | — | 83.4 | 43.5 | 87.3 | — | — |
| MoCo.+Mix. | 91.5 | 62.7 | 90.1 | — | 83.2 | 43.2 | 88.4 | — | — |
| SimC.+Dif. | 91.3 | 62.8 | 90.6 | 46.3 | 84.2 | 46.3 | 80.1 | 19.7 | 63.8 |
| MoCo.+Dif. | 91.4 | 64.1 | 90.5 | 46.2 | 84.1 | 47.9 | 83.6 | 19.9 | 65.3 |
| DiffAug | **93.4**(+1.9) | **69.9**(+3.8) | **92.5**(+1.9) | **49.7**(+3.4) | **86.2**(+2.0) | **48.6**(+0.7) | **89.4**(+1.0) | **19.9**(+0.0) | **67.8**(+2.5) |

## 3.1 DIFFAUG OUTPERFORMS HAND DESIGNED AUGMENTATIONS ON VISION DATASETS

We showcase DiffAug's versatility across multiple test protocols, emphasizing its potential to enhance vision data. We focus on data augmentation techniques that can be used for unsupervised contrastive learning in the unsupervised case. Therefore, the comparison does not include some labeling-based methods (Nichol et al., 2021; He et al., 2023; Trabucco et al., 2023; Zhang et al., 2023a). Comparative outcomes based on linear-test performance and clustering are detailed in Table 1, and the data of CF10 and CF100 is from Huang et al. (2023). Different baseline methods use different hand-designed data augmentations (in Appendix C.3).

**Test Protocols.** Experiments are performed on CIFAR-10 [CF10] and CIFAR-100 [CF100] (Krizhevsky et al., 2009), STL10 (Coates et al., 2011), TinyImageNet [TINet] (Le & Yang, 2015) dataset. We followed a procedure similar to SimCLR (Chen et al., 2020) for the Linear-test performance assessment. We evaluated the model's representations linearly on top of the frozen features. This ensures that the quality of the representations is attributed only to the pre-training task, without any influence from potential fine-tuning. We used the ResNet (He et al., 2015) backbone from the baseline. In contrast, for DiffAug, its semantic encoder served as the contrastive learning backbone, trained using DiffAug-augmented images. We extracted feature vectors from the models for the K-means clustering evaluation, leaving out the top classification

Table 2: **Comparison of the classification performance on biological dataset.** Unsupervised representation learning methods that have been widely used on biological analyze are compared, including kPCA (Halko et al., 2010), Ivis (Szubert et al., 2019), PHATE (Moon & van Dijk, 2019), PUMAP (Sainburg et al., 2021), PaCMAP (Wang et al., 2022), EVNet (Zang et al., 2022a) and hNNE (Sarfraz et al., 2022). The improvement over the best baseline is shown in parentheses.

| | kPCA | PUMAP | Ivis | PHATE | Topo-AE | PaCMAP | EVNet | hNNE | DiffAug |
|---|---|---|---|---|---|---|---|---|---|
| GA1457 | 40.2 | 66.0 | 36.9 | 72.2 | 74.6 | 85.3 | 85.7 | 77.4 | **92.7**(+7.0) |
| SAM561 | 34.6 | 59.9 | 45.6 | 71.5 | 72.4 | 83.7 | 83.6 | 83.8 | **89.3**(+4.5) |
| MC1374 | 45.6 | 62.2 | 45.8 | 61.3 | 61.3 | 61.3 | 71.4 | 62.3 | **71.8**(+0.4) |
| HCL500 | 26.5 | 36.3 | 24.4 | 33.8 | 56.0 | 36.2 | 62.3 | 62.2 | **64.7**(+2.4) |
| AVE | 36.7 | 56.1 | 38.2 | 59.1 | 66.1 | 74.8 | 75.8 | 71.4 | **79.6**(+3.8) |

layer. We then applied K-means clustering to these features. The primary metric for evaluation was clustering accuracy. Details of the experimental setup are in Appendix C. The training strategy of DiffAug is E-step: 200 epochs → M-step: 400 epoch → E-step: 800 epoch. The data of training time consumption is in the Table A.3.

**Analysis.** From Table 1, it's evident that DiffAug consistently outperforms state-of-the-art (SOTA) methods across all datasets. It surpasses other techniques by at least 1.0% in five out of the eight projects, raising the average metrics by a minimum of 2.5% compared to other evaluated methods. This showcases the effectiveness of DiffAug's data augmentation. (a) *Beyond hand-designed augmentation methods.* DiffAug's versatility indicates that its approach is on par with, or even better than, traditional hand-crafted methods. The encoder in DiffAug produces robust features. (b) *Beyond Mixup improved contrastive learning methods.* DiffAug outperforms the Mixup improved contrastive learning method of typical contrast learning methods, and additionally, models trained using DiffAug-generated data and contrast learning methods bring some improvement. (c) For datasets with many classes, like CF100 and TINet, DiffAug's encoder might only sometimes capture every detail. Still, augmented data is crucial in guiding contrastive learning to produce better results.

## 3.2 DIFFAUG DELIVERS A BOOST IN UNSUPERVISED REPRESENTATION LEARNING FOR HIGH-COST BIOLOGICAL DATA

Next, we benchmark DiffAug against SOTA unsupervised representation learning methods in biological datasets. We present the comparative outcomes of SVC classification performance (Platt, 1999) in Table 2. The data of training time consumption is in the Table A.5.

**Test protocols.** Experiments are performed on biological datasets, including GA1457 (Rouillard et al., 2016), SAM (Weber & Robinson, 2016), MC1374 (Han et al., 2018), and HCL500 (Han et al., 2020) datasets. To assess the efficacy of the proposed methods, following Wang et al. (2022); Sarfraz et al. (2022), we utilized linear SVM performance to evaluate the performance of different methods. In the linear SVM evaluation, embeddings are partitioned with 90% designated for training and 10% for testing. Detailed specifics of this configuration are elaborated in the Appendix D. The training strategy of DiffAug is E-step: 330 epochs → M-step: 330 epoch → E-step: 340 epoch. The training details and the correctness change curve are in Fig. A.2.

**Analysis.** DiffAug consistently surpasses all other methods across eight evaluations spanning four datasets, registering a performance enhancement between 0.4% and 7.0% over its counterparts. Several key advantages of DiffAug emerge, especially when considering classification metrics: (a) Notably, the strengths of DiffAug aren't confined to vision data. It also excels in areas such as biology, where data is difficult to visualize and understand by humans and where it is challenging to manually design appropriate data augmentation methods. (b) Data processed through DiffAug exhibits reduced overlap among distinct groups, facilitating enhanced classification and clustering. This suggests that DiffAug delineates more explicit boundaries between data categories, culminating in more precise outcomes. (c) The approach underpinning DiffAug is versatile, making it a valuable addition to other unsupervised learning techniques. Historically, the quest for potent data augmentation strategies in biology has been arduous. We posit that DiffAug paves new avenues in this domain.

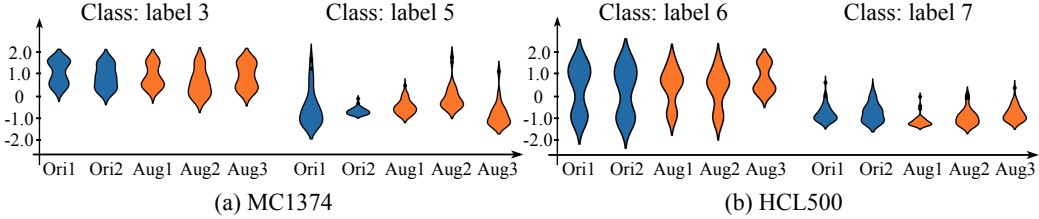

(a) CF10        (b) STL10

Figure 3: **The display of original and generated images illustrates that DiffAug generates semantically similar augmented images.** Ori means original image and Aug1, Aug2 and Aug3 are augmented images. More detailed results are in the appendix.

(a) MC1374        (b) HCL500

Figure 4: **The violin plot of original and generated samples illustrates that the DiffAug generated samples have a similar distribution to the original samples.** Ori1 and Ori2 are two original sample and Aug1, Aug2 and Aug3 are augmented data. Due to the large number of zero values in the biological data, we chose the genes with the highest 100 expression to plot violin plots to show the data distribution. Class: label 3 and Class: label 5 indicate two different classes in the dataset.

### 3.3 DIFFAUG EFFECTIVENESS ANALYSIS

Next, we verify the tasks and roles of each module by showing the details of how DiffAug works.

**Effectiveness analysis of diffusion generator.** The diffusion module generates new positive data by inputting the provided condition vector. To demonstrate that the diffusion module works appropriately, we show the generation results for both the image and biological datasets (in Fig. 3 and Fig. 4). A more detailed implementation and more results are in the Appendix C and Appendix D. We can observe that the generated data retains semantic similarity to the original data. For example, the objects described in the image data are consistent, while the distribution of the gene is also consistent. At the same time, the generated data is not simply copied but varied without changing the semantic information.

In addition, to further explore the semantic differences between the newly generated and original data, we computed the cos-similarity of the original augmented sample in latent space. As depicted in Fig. 5, DiffAug's similarity distribution is smoother and broader. In comparison, Mixup tends to produce augmentations that are very similar semantically, while methods like cropping might introduce data with semantically distinct noise samples.

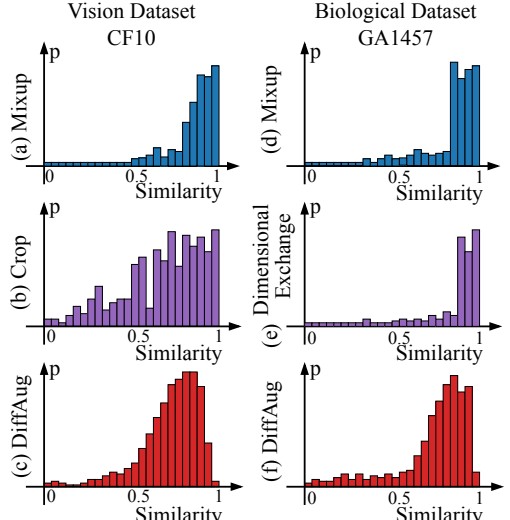

Figure 5: **Hist plot of the cosine similarity between original data and the augmentation data in latent space indicate that DiffAug generates semantically smooth augmentations.** For the image data, we compared similar mixups with random cropping. For biological datasets, we compared same-label Mixup and random dimension swapping.

**Effectiveness analysis of semantic encoder.** Next, we confirm that the semantic encoder of DiffAug works well by visualizing the representation of DiffAug and baseline methods (in Fig. 6). The

Table 3: **Ablation study of the semantic encoder includes DiffAug's encoder is necessary and can efficiently generate conditional vectors.** Linear-tests performance of different ablation setups on on vision dataset and biological dataset.

| Datasets | Vision Datasets | | | | Biological Datasets | | | |
|---|---|---|---|---|---|---|---|---|
| | CF10 | CF100 | STL10 | TINet | GA1457 | SAM561 | MC1374 | HCL500 |
| **A1.** Gen$(\cdot)$ + Sup. Condition | **93.4** | **70.9** | **92.9** | 45.9 | 92.5 | **89.6** | 71.1 | 63.9 |
| **A2.** Gen$(\cdot)$ + Rand. Condition | 34.2 | 10.4 | 30.1 | 7.3 | 10.5 | 16.9 | 13.9 | 10.0 |
| **A3.** Gen$(\cdot)$ + Enc$(\cdot\|\theta)$ (DiffAug) | **93.4** | 69.9 | 92.5 | **49.7** | 92.7 | 88.3 | **71.8** | **64.7** |

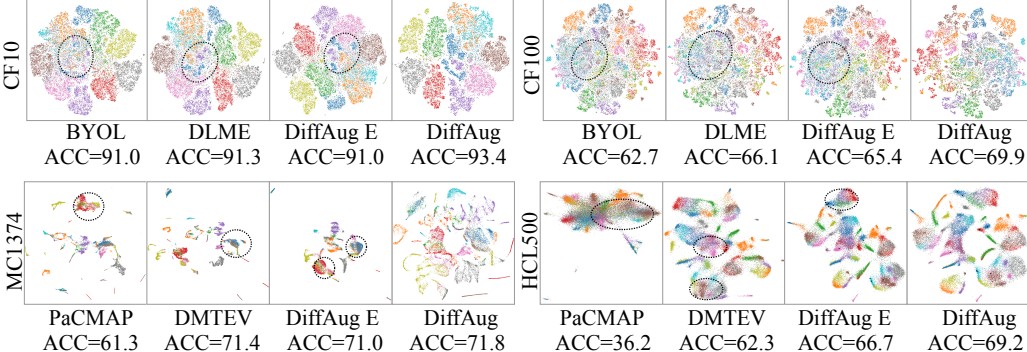

Figure 6: **The scatter visualization of representation indicates DiffAug's encoder learns cleaner embedding.** The colors represent different categories; there are 100 categories in CF100; we used the superclasses label provided by Deng et al. (2021). DiffAug E means only using the soft contrastive loss to train the encoder.

t-SNE (Van der Maaten & Hinton, 2008) is used to analyze the BYOL, DLME, and DiffAug embedding on CF10, CF100, MC1374, and HCL500 datasets. The results show that DiffAug's encoder learns cleaner embedding than baseline methods. In Fig. 6, DiffAug E means the first E-step's results of the DiffAug. By comparing DiffAug E and DiffAug, we observe that the augmented data further improves the embedding quality, significantly enhancing the depiction of local structures. The same conclusion is shown in Fig. A.2.

## 3.4 ABLATION STUDY AND EFFECTIVENESS OF EACH COMPONENT

**Ablation study of the semantic encoder.** In the ablation study presented in Table. 3, we consider three configurations: A1 and A2 confirm the significance of DiffAug's semantic encoder by ablating it in two ways. A1 directly uses supervised one hot label as the conditional, bypassing the condition vectors generated by the unsupervised neural network. A2 employs random conditional vectors instead of those the encoder produces. A3 means the proposed DiffAug method. The results from these experiments can be found in Table 3. We observe that the average performance of A1 is highest due to the access to the label. And not accessing the label at all brings a huge performance drop. The results in A3 illustrate that DiffAug's performance is comparable to the fully supervised condition, demonstrating its ability to model supervised annotation within an unsupervised framework.

**Ablation study of training strategy and scl loss function.** For Ablation in Table. 4, B1 means that the model is trained by SimCLR (Chen et al., 2020). B2 omits the diffusion loss and trains the encoder with only the soft contrastive learning loss. B3 omits the soft contrastive learning loss and trains the encoder with InfoNCE loss. B4 and B5 talk about the training strategy of DiffAug. B4 denotes training the model by integrating two loss functions, i.e., mixing E-Step and M-Step to update the parameters of both networks simultaneously through a single forward propagation.B5 denotes the default training strategy, which trains the model by alternating the two loss functions. The results from these experiments can be found in Table 4. First, we observe that either replacing the scl loss or replacing the diff model (B2 or B3) brings about performance degradation, which implies that the two modules of DiffAug work in conjunction with each other. Second, we observe

Table 4: **Ablation study of scl loss function and training strategy.** The classifier accuracy of each setting is displayed in this table. Soft contrastive learning is improved with typical contrast learning, and EM training is more stable.

| Datasets | Vision Datasets | | | | Biological Datasets | | | |
|---|---|---|---|---|---|---|---|---|
| | CF10 | CF100 | STL10 | TINet | GA1457 | SAM561 | MC1374 | HCL500 |
| **B1.** SimCLR | 89.8 | 57.4 | 86.9 | 38.4 | 9.4 | 16.8 | 14.3 | 16.8 |
| **B2.** DiffAug w/o $\mathcal{L}_{df}$ | 91.3 | 66.1 | 90.1 | 44.9 | 89.1 | 82.1 | 59.3 | 62.3 |
| **B3.** DiffAug w/o $\mathcal{L}_{scl}$ | 92.7 | 68.4 | 90.9 | 45.1 | 89.2 | 82.4 | 69.2 | 61.3 |
| **B4.** DiffAug Syn. Training | 92.9 | 69.7 | 92.7 | 45.3 | 90.1 | **89.6** | 68.1 | 62.3 |
| **B5.** DiffAug EM Training | **93.4** | **69.9** | 92.5 | **49.7** | **92.7** | 88.3 | **71.8** | **64.7** |

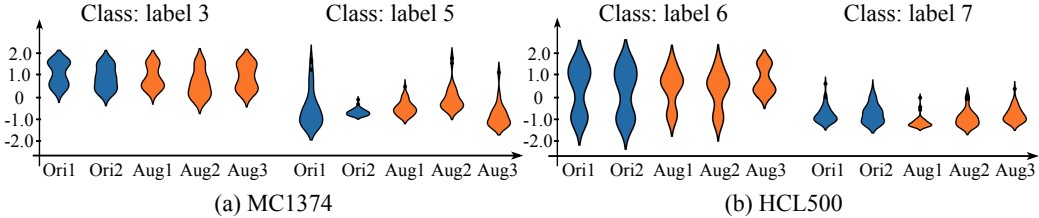

Figure 7: **Hypter-parametric $\lambda$ analysis.** The Box plots of augmentation strength hyperparameters $\lambda$ and model Linear-tests performance. For each parameter, statistical results of experiments based on grid search were performed.

that on some datasets, the performance of the two training strategies (B4 and B5) is comparable, but on others, the EM method demonstrates higher stability. We attribute this to the fact that the difficulty of diffusion model training varies from data to data, and simultaneous training may result in the two modules being unable to match at all times, bringing about instability in training. However, the E-M training approach avoids this problem.

### 3.5 HYPERPARAMETRIC ANALYSIS AND THE TOXICITY OF GENERATED DATA

Finally, we investigate the performance improvement and potential toxicity of the DiffAug method through hyperparametric analysis. The hyperparameter $\lambda$ determines how often the model generated by DiffAug affects the training of the semantic encoder. Introducing the least amount of augmentation data ($\lambda = 0$) brings the method back to traditional contrastive learning methods, while too much ($\lambda = 1$) will lead to encoder to crash. To demonstrate this, we tested the model performance of different $\lambda$ counterparts on two visual datasets (CF10, CF100) and two biological datasets (SAM561 and MC1374). As shown in Fig. 7, the change in performance brought about by $\lambda$ is consistent across datasets. Specifically, setting $\lambda = 0.1$ or $\lambda = 0.15$ provides the most significant gain. We believe that $\lambda = 0.1$ may be a suitable default setting for most datasets.

## 4 CONCLUSION

In summary, we presented DiffAug, an innovative contrastive learning framework that leverages diffusion-based augmentation to enhance the robustness and generalization of unsupervised learning. Unlike many existing methods, DiffAug operates independently of prior knowledge or external labels, positioning it as a versatile augmentation tool with notable performance in vision and life sciences. Our tests reveal that DiffAug consistently boosts classification and clustering accuracy across multiple datasets, such as CF10, CF100, STL10, TINet, HCL500, GA1457, SAM561, and MC1374. Given its capabilities, we see DiffAug evolving into a benchmark data augmentation method in unsupervised contrastive learning. We suggest fellow researchers delve into DiffAug and harness its potential for diverse applications.

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

Appendix:
Boosting Unsupervised Contrastive Learning
Using Diffusion-Based Data Augmentation From Scratch

# Contents

## A    APPENDIX: RELATED WORKS

**Generative Models**    Generative models have been the subject of growing interest and rapid advancement. Earlier methods, including VAEs (Kingma & Welling, 2014) and GANs (Goodfellow et al., 2014), showed initial promise generating realistic images, and were scaled up in terms of resolution and sample quality (Brock et al., 2019; Razavi et al., 2019). Despite the power of these methods, many recent successes in photorealistic image generation were the result of diffusion models (Ho et al., 2020; Nichol & Dhariwal, 2021; Saharia et al., 2022; Nichol et al., 2022; Ramesh et al., 2022). Diffusion models have been shown to generate higher-quality samples compared to their GAN counterparts (Dhariwal & Nichol, 2021), and developments like classifier free guidance (Ho & Salimans, 2022) have made text-to-image generation possible. Recent emphasis has been on training these models with internet-scale datasets like LAION-5B (Schuhmann et al., 2022). Generative models trained at internet-scale (Rombach et al., 2022; Saharia et al., 2022; Nichol et al., 2022; Ramesh et al., 2022) have unlocked several application areas where photorealistic generation is crucial.

**Synthetic Image Data Generation**    Training neural networks on synthetic data from generative models was popularized using GANs (Antoniou et al., 2017; Tran et al., 2017; Zheng et al., 2017).

Various applications for synthetic data generated from GANs have been studied, including representation learning (Jahanian et al., 2022), inverse graphics (Zhang et al., 2021a), semantic segmentation (Zhang et al., 2021b), and training classifiers (Tanaka & Aranha, 2019; Dat et al., 2019; Yamaguchi et al., 2020; Besnier et al., 2020; Xiong et al., 2020; Wickramaratne & Mahmud, 2021; Haque, 2021). More recently, synthetic data from diffusion models has also been studied in a few-shot setting (He et al., 2022). These works use generative models that have likely seen images of target classes and, to the best of our knowledge, we present the first analysis for synthetic data on previously unseen concepts. Du et al. (2023) proposed the DREAM-OOD framework, which uses diffusion models to generate photo-realistic outliers from in-distribution data for improved OOD detection. By learning a text-conditioned latent space, it visualizes imagined outliers directly in pixel space, showing promising results in empirical studies. Zhang et al. (2023a) developed the Guided Imagination Framework (GIF) using generative models like DALL-E2 and Stable Diffusion for dataset expansion, enhancing accuracy in both natural and medical image datasets.

**Synthetic Biology Data Generation** The realm of synthetic biology has witnessed a surge in the utilization of data-driven approaches, particularly with the advent of advanced computational models. The generation of synthetic biological data has been instrumental in predicting protein structures (McGibbon et al., 2023). The use of Generative Adversarial Networks (GANs) has also found its way into this domain, aiding in the c reation of synthetic DNA sequences (Zheng et al., 2023; Li & Zhang, 2022; Han et al., 2019) and simulating cell behaviors (Botton et al., 2022). Furthermore, the integration of machine learning with synthetic biology has paved the way for innovative solutions in drug discovery (Blanco-Gonzalez et al., 2023; McGibbon et al., 2023). Unlike the synthetic image data generation, where models have often seen images of target classes, synthetic biology data generation often grapples with the challenge of generating data for entirely novel biological entities. This presents a unique set of challenges and opportunities, pushing the boundaries of what synthetic data can achieve in the realm of biology.

## B  APPENDIX: DETAILS OF CONTRASTIVE LEARNING AND SOFT CONTRASTIVE LEARNING

### B.1  THE T-KERNEL SIMILARITY IN SOFT CONTRASTIVE LEARNING

To map the high-dimensional embedding vector to a probability value, a kernel function $\mathcal{S}(\cdot)$ is used. In this paper, we use the t-distribution kernel function $\mathcal{S}^\nu(\cdot)$ because it exposes the degrees of freedom and allows us to adjust the closeness of the distribution in the dimensionality reduction mapping (Li et al., 2021). The t-distribution kernel function is defined as follows,

$$\mathcal{S}(\mathbf{z}_i, \mathbf{z}_j) = \Gamma\left((\nu+1)/2\right)\left(1 + \|z_i - z_j\|_2^2/\nu\right)^{-\frac{\nu+1}{2}}/\sqrt{\nu\pi}\Gamma\left(\nu/2\right), \tag{9}$$

where $\Gamma(\cdot)$ is the Gamma function. The degrees of freedom $\nu$ control the shape of the kernel function. The different degrees of freedom $(\nu^y, \nu^z)$ is used in $\mathcal{R}^y$ and $\mathcal{R}^z$ for the dimensional reduction mapping.

### B.2  WHY SOFT CONTRASTIVE LEARNING IS A SOFTENED VERSION OF CONTRASTIVE LEARNING

**Lemma 1.** *Let* $\mathcal{L}_{\text{cl}} = -\log \mathcal{Q}\left(\mathbf{z}_i, \mathbf{z}_i^+\right) + \log\left[\mathcal{Q}\left(\mathbf{z}_i, \mathbf{z}_i^+\right) + \sum_{\mathbf{z}_i^- \in V^-} \mathcal{Q}\left(\mathbf{z}_i, \mathbf{z}_i^-\right)\right]$ *and* $\mathcal{L}_{\text{cl}}^p = -\sum_{j=1}^{N_K+1}\left\{\mathcal{H}_{ij}\log Q_{ij} + (1 - \mathcal{H}_{ij})\log \dot{Q}_{ij}\right\}$ *Then* $\lim_{x\to\infty}\mathcal{L}_{\text{cl}} - \mathcal{L}_{\text{cl}}^p = 0$.

*Proof.* We start with $L_{\text{CL}} = -\log \frac{\exp(S(z_i,z_j))}{\sum_{k=1}^{N_K}\exp(S(z_i,z_k))}$ (Eq. (3)), then

$$L_{\text{CL}} = \log N_K - \log \frac{\exp(S(z_i, z_j))}{\frac{1}{N_K}\sum_{k=1}^{N_K}\exp(S(z_i, z_k))}.$$

We are only concerned with the second term that has the gradient. Let $(i, j)$ are positive pair and $(i, k_1), \cdots, (i, k_N)$ are negative pairs. The overall loss associated with point $i$ is:

$$
- \log \frac{\exp(S(z_i, z_j))}{\frac{1}{N_K} \sum_{k=1}^{N_K} \exp(S(z_i, z_k))}
$$

$$
= - \left[ \log \exp(S(z_i, z_j)) - \log \frac{1}{N_K} \sum_{k=1}^{N_K} \exp(S(z_i, z_k)) \right]
$$

$$
= - \left[ \log \exp(S(z_i, z_j)) - \sum_{k=1}^{N_K} \log \exp(S(z_i, z_k)) + \sum_{k=1}^{N_K} \log \exp(S(z_i, z_k)) - \log \frac{1}{N_K} \sum_{k=1}^{N_K} \exp(S(z_i, z_k)) \right]
$$

$$
= - \left[ \log \exp(S(z_i, z_j)) - \sum_{k=1}^{N_K} \log \exp(S(z_i, z_k)) + \log \Pi_{k=1}^{N_K} \exp(S(z_i, z_k)) - \log \frac{1}{N_K} \sum_{k=1}^{N_K} \exp(S(z_i, z_k)) \right]
$$

$$
= - \left[ \log \exp(S(z_i, z_j)) - \sum_{k=1}^{N_K} \log \exp(S(z_i, z_k)) + \log \frac{\Pi_{k=1}^{N_K} \exp(S(z_i, z_k))}{\frac{1}{N_K} \sum_{k=1}^{N_K} \exp(S(z_i, z_k))} \right]
$$

We focus on the case where the similarity is normalized, $S(z_i, z_k) \in [0, 1]$. The data $i$ and data $k$ is the negative samples, then $S(z_i, z_k)$ is near to 0, $\exp(S(z_i, z_k))$ is near to 1, thus the $\frac{\Pi_{k=1}^{N_K} \exp(S(z_i, z_k))}{\frac{1}{N} \sum_{k=1}^{N_K} \exp(S(z_i, z_k))}$ is near to 1, and $\log \frac{\Pi_{k=1}^{N_K} \exp(S(z_i, z_k))}{\frac{1}{N} \sum_{k=1}^{N_K} \exp(S(z_i, z_k))}$ near to 0. We have

$$
L_{\mathrm{CL}} \approx - \left[ \log \exp(S(z_i, z_j)) - \sum_{k=1}^{N_K} \log \exp(S(z_i, z_k)) \right]
$$

We denote $ij$ and $ik$ by a uniform index and use $\mathcal{H}_{ij}$ to denote the homology relation of $ij$.

$$
L_{\mathrm{CL}} \approx - \left[ \log \exp(S(z_i, z_j)) - \sum_{k=1}^{N_K} \log \exp(S(z_i, z_k)) \right]
$$

$$
\approx - \left[ \mathcal{H}_{ij} \log \exp(S(z_i, z_j)) - \sum_{j=1}^{N_K} (1 - \mathcal{H}_{ij}) \log \exp(S(z_i, z_j)) \right]
$$

$$
\approx - \left[ \sum_{j=1}^{N_K+1} \{ \mathcal{H}_{ij} \log \exp(S(z_i, z_j)) + (1 - \mathcal{H}_{ij}) \log \{ \exp(-S(z_i, z_j)) \} \} \right]
$$

we define the similarity of data $i$ and data $j$ as $Q_{ij} = \exp(S(z_i, z_j))$ and the dissimilarity of data $i$ and data $j$ as $\dot{Q}_{ij} = \exp(-S(z_i, z_j))$.

$$
L_{\mathrm{CL}} \approx - \left[ \sum_{j=1}^{N_K+1} \left\{ \mathcal{H}_{ij} \log Q_{ij} + (1 - \mathcal{H}_{ij}) \log \dot{Q}_{ij} \right\} \right]
$$

$\square$

**The proposed SCL loss is a smoother CL loss:**

This proof tries to indicate that the proposed SCL loss is a smoother CL loss. We discuss the differences by comparing the two losses to prove this point. the forward propagation of the network is, $z_i = H(\hat{z}_i), \hat{z}_i = F(x_i), z_j = H(\hat{z}_j), \hat{z}_j = F(x_j)$. We found that we mix $y$ and $\hat{z}$ in the main text, and we will correct this in the new version. So, in this section $z_i = H(y_i), y_i = F(x_i), z_j = H(y_j), y_j = F(x_j)$ is also correct.

Let $H(\cdot)$ satisfy $K$-Lipschitz continuity, then $d_{ij}^z = k^* d_{ij}^y, k^* \in [1/K, K]$, where $k^*$ is a Lipschitz constant. The difference between $L_{\text{SCL}}$ loss and $L_{\text{CL}}$ loss is,

$$L_{\text{CL}} - L_{\text{SCL}} \approx \sum_j \left[ \left( \mathcal{H}_{ij} - [1 + (e^\alpha - 1)\mathcal{H}_{ij}]\kappa\left(d_{ij}^y\right) \right) \log\left( \frac{1}{\kappa\left(d_{ij}^z\right)} - 1 \right) \right]. \tag{10}$$

Because the $\alpha > 0$, the proposed SCL loss is the soft version of the CL loss. if $\mathcal{H}_{ij} = 1$, we have:

$$(L_{\text{CL}} - L_{\text{SCL}})|_{\mathcal{H}_{ij}=1} = \sum \left[ \left( (1 - e^\alpha)\kappa\left(k^* d_{ij}^z\right) \right) \log\left( \frac{1}{\kappa\left(d_{ij}^z\right)} - 1 \right) \right] \tag{11}$$

then:

$$\lim_{\alpha \to 0} (L_{\text{CL}} - L_{\text{SCL}})|_{\mathcal{H}_{ij}=1} = \lim_{\alpha \to 0} \sum \left[ \left( (1 - e^\alpha)\kappa\left(k^* d_{ij}^z\right) \right) \log\left( \frac{1}{\kappa\left(d_{ij}^z\right)} - 1 \right) \right] = 0 \tag{12}$$

Based on Eq.(12), we find that if $i, j$ is neighbor ($\mathcal{H}_{ij} = 1$) and $\alpha \to 0$, there is no difference between the CL loss $L_{\text{CL}}$ and SCL loss $L_{\text{SCL}}$. When if $\mathcal{H}_{ij} = 0$, the difference between the loss functions will be the function of $d_{ij}^z$. The CL loss $L_{\text{CL}}$ only minimizes the distance between adjacent nodes and does not maintain any structural information. The proposed SCL loss considers the knowledge both comes from the output of the current bottleneck and data augmentation, thus less affected by view noise.

**Details of Eq. (10).** Due to the very similar gradient direction, we assume $\dot{Q}_{ij} = 1 - Q_{ij}$. The contrastive learning loss is written as,

$$L_{\text{CL}} \approx -\sum \left\{ \mathcal{H}_{ij} \log Q_{ij} + (1 - \mathcal{H}_{ij}) \log\left(1 - Q_{ij}\right) \right\} \tag{13}$$

where $\mathcal{H}_{ij}$ indicates whether $i$ and $j$ are augmented from the same original data.

The SCL loss is written as:

$$L_{\text{SCL}} = -\sum \left\{ P_{ij} \log Q_{ij} + (1 - P_{ij}) \log\left(1 - Q_{ij}\right) \right\} \tag{14}$$

According to Eq. (4) and Eq. (5), we have

$$P_{ij} = R_{ij}\kappa(d_{ij}^y) = R_{ij}\kappa(y_i, y_j), R_{ij} = \begin{cases} e^\alpha & \text{if } \mathcal{H}(x_i, x_j) = 1 \\ 1 & \text{otherwise} \end{cases},$$
$$Q_{ij} = \kappa(d_{ij}^z) = \kappa(z_i, z_j), \tag{15}$$

For ease of writing, we use distance as the independent variable, $d_{ij}^y = \|y_i - y_j\|_2, d_{ij}^z = \|z_i - z_j\|_2$.

The difference between the two loss functions is:

$$L_{\mathrm{CL}} - L_{\mathrm{SCL}}$$

$$= -\sum \left[ \mathcal{H}_{ij} \log \kappa \left(d_{ij}^z\right) + (1 - \mathcal{H}_{ij}) \log \left(1 - \kappa \left(d_{ij}^z\right)\right) - R_{ij} \kappa \left(d_{ij}^y\right) \log \kappa \left(d_{ij}^z\right) - \left(1 - R_{ij} \kappa \left(d_{ij}^y\right)\right) \log \left(1 - \kappa \left(d_{ij}^z\right)\right) \right]$$

$$= -\sum \left[ \left(\mathcal{H}_{ij} - R_{ij} \kappa \left(d_{ij}^y\right)\right) \log \kappa \left(d_{ij}^z\right) + \left(1 - \mathcal{H}_{ij} - 1 + R_{ij} \kappa \left(d_{ij}^y\right)\right) \log \left(1 - \kappa \left(d_{ij}^z\right)\right) \right]$$

$$= -\sum \left[ \left(\mathcal{H}_{ij} - R_{ij} \kappa \left(d_{ij}^y\right)\right) \log \kappa \left(d_{ij}^z\right) + \left(R_{ij} \kappa \left(d_{ij}^y\right) - \mathcal{H}_{ij}\right) \log \left(1 - \kappa \left(d_{ij}^z\right)\right) \right]$$

$$= -\sum \left[ \left(\mathcal{H}_{ij} - R_{ij} \kappa \left(d_{ij}^y\right)\right) \left(\log \kappa \left(d_{ij}^z\right) - \log \left(1 - \kappa \left(d_{ij}^z\right)\right)\right) \right]$$

$$= \sum \left[ \left(\mathcal{H}_{ij} - R_{ij} \kappa \left(d_{ij}^y\right)\right) \log \left( \frac{1}{\kappa \left(d_{ij}^z\right)} - 1 \right) \right] \tag{16}$$

Substituting the relationship between $\mathcal{H}_{ij}$ and $R_{ij}$, $R_{ij} = 1 + (e^\alpha - 1)\mathcal{H}_{ij}$, we have

$$L_{\mathrm{CL}} - L_{\mathrm{SCL}} = \sum \left[ \left(\mathcal{H}_{ij} - [1 + (e^\alpha - 1)\mathcal{H}_{ij}]\kappa \left(d_{ij}^y\right)\right) \log \left( \frac{1}{\kappa \left(d_{ij}^z\right)} - 1 \right) \right] \tag{17}$$

We assume that network $H(\cdot)$ to be a Lipschitz continuity function, then

$$\frac{1}{K} H(d_{ij}^z) \leq d_{ij}^y \leq K H(d_{ij}^z) \quad \forall i, j \in \{1, 2, \cdots, N\} \tag{18}$$

We construct the inverse mapping of $H(\cdot)$ to $H^{-1}(\cdot)$,

$$\frac{1}{K} d_{ij}^z \leq d_{ij}^y \leq K d_{ij}^z \quad \forall i, j \in \{1, 2, \cdots, N\} \tag{19}$$

and then there exists $k^*$:

$$d_{ij}^y = k^* d_{ij}^z \quad k^* \in [1/K, K] \quad \forall i, j \in \{1, 2, \cdots, N\} \tag{20}$$

Substituting the Eq.(20) into Eq.(17).

$$L_{\mathrm{CL}} - L_{\mathrm{SCL}} = \sum \left[ \left(\mathcal{H}_{ij} - [1 + (e^\alpha - 1)\mathcal{H}_{ij}]\kappa \left(k^* d_{ij}^z\right)\right) \log \left( \frac{1}{\kappa \left(d_{ij}^z\right)} - 1 \right) \right] \tag{21}$$

## C  APPENDIX: DETAILS OF VISION EXPERIMENTS

### C.1  DATASET SETUPS

Experiments are performed on CIFAR-10 [CF10][1] and CIFAR-100[2] [CF100] (Krizhevsky et al., 2009), STL10[3] (Coates et al., 2011), TinyImageNet[4] [TINet] (Le & Yang, 2015) dataset.

To compare with the two different baseline methods, the setting of the dataset is shown in Table. A.1.

---

[1]https://www.cs.toronto.edu/ kriz/cifar.html

[2]https://www.cs.toronto.edu/ kriz/cifar.html

[3]https://cs.stanford.edu/ acoates/stl10/

[4]https://www.kaggle.com/c/tiny-imagenet

Table A.1: Dataset setting of linear-test Performance.

| Dataset | Train Split | Test Split | Train Samples | Test Samples | Classes |
|---------|-------------|------------|---------------|--------------|---------|
| CF10 | Train | Test | 50,000 | 10,000 | 10 |
| CF100 | Train | Test | 50,000 | 10,000 | 100 |
| STL10 | Train + Unlabeled | Test | 5,000+100,000 | 8,000 | 10 |
| TINet | Train | Test | 100,000 | 100,000 | 200 |

Table A.2: Dataset setting of clustering test.

| Dataset | Train & Test Split | Train & Test Samples | Classes |
|---------|--------------------|-----------------------|---------|
| CF10 | Train+Test | 60,000 | 10 |
| CF100 | Train+Test | 60,000 | 20 |
| STL10 | Train+Test | 13,000 | 10 |
| TIN | Train | 100,000 | 200 |

## C.2 BASELINE METHODS AND IMPLEMENTATION DETAILS

The contrastive learning methods, including SimCLR (Chen et al., 2020), MOCO v2 (He et al., 2020), BYOL (Grill et al., 2020), SimSiam (Chen & He, 2021), and DLME (Zang et al., 2022b) are chosen for comparison. The SimC.+Mix. and MoCo.+Mix. are SimCLR and MoCoV2 with Mixup data augmentation which processed by Zhang et al. (2022). The SimC.+Dif. and MoCo.+Mix. are SimCLR and MoCoV2 with DiffAug data augmentation. Improvements over the best baseline are shown in parentheses.

For the Linear-test performance assessment, we followed a procedure similar to SimCLR (Chen et al., 2020). We evaluated the model's representations linearly on top of the frozen features. This ensures that the quality of the representations is attributed only to the pre-training task, without any influence from potential fine-tuning. We used the ResNet-50 (He et al., 2015) backbone as the encoder and a standard diffusion backbone as diffusion model (in code below). In contrast, for DiffAug, its semantic encoder served as the contrastive learning backbone, trained using DiffAug-augmented images. For the kMeans clustering evaluation, we extracted feature vectors from the models, leaving out the top classification layer. We then applied kMeans clustering to these features. The primary metric for evaluation was clustering accuracy.

Listing 1: DiffusionModel for Vision Task

```
1   class DiffusionModelVision(nn.Module):
2       def __init__(self, c_in=3, c_out=3, time_dim=256):
3           super().__init__()
4           self.time_dim = time_dim
5           self.remove_deep_conv = remove_deep_conv
6           self.inc = DoubleConv(c_in, 16)
7           self.down1 = Down(16, 32)
8           self.sa1 = SelfAttention(32)
9           self.down2 = Down(32, 64)
10          self.sa2 = SelfAttention(64)
11          self.down3 = Down(64, 64)
12          self.sa3 = SelfAttention(64)
13          self.up1 = Up(128, 32)
14          self.sa4 = SelfAttention(32)
15          self.up2 = Up(64, 16)
16          self.sa5 = SelfAttention(16)
17          self.up3 = Up(32, 16)
18          self.sa6 = SelfAttention(16)
19          self.outc = nn.Conv2d(16, c_out, kernel_size=1)
20          def pos_encoding(self, t, channels):
21          inv_freq = 1.0 / (10000 ** (torch.arange(0, channels, 2,
                    device=one_param(self).device).float() / channels))
22          pos_enc_a = torch.sin(t.repeat(1, channels // 2) * inv_freq)
23          pos_enc_b = torch.cos(t.repeat(1, channels // 2) * inv_freq)
24          pos_enc = torch.cat([pos_enc_a, pos_enc_b], dim=-1)
```

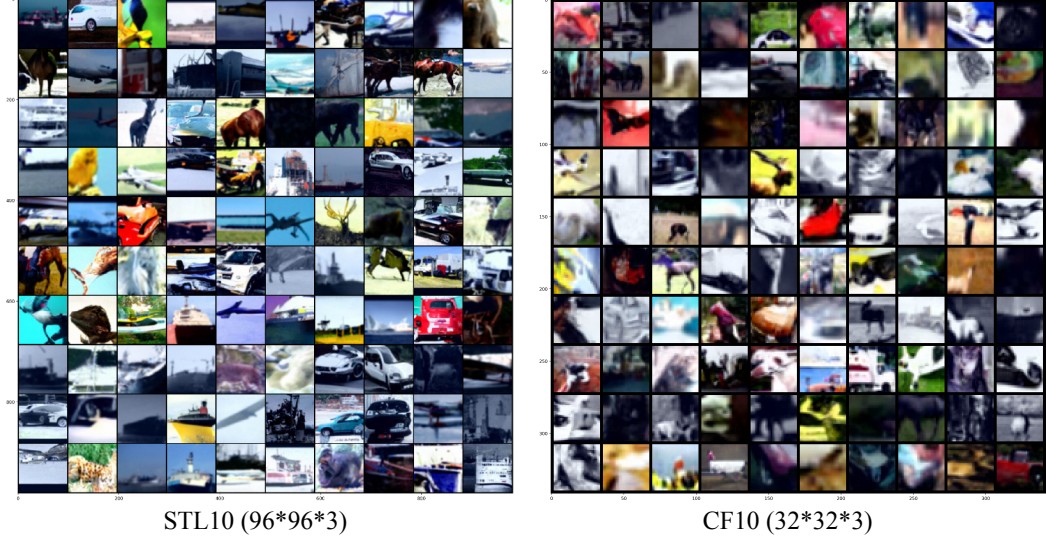

STL10 (96*96*3)                    CF10 (32*32*3)

Figure A.1: **The display of original and generated images illustrates that DiffAug generates semantically similar augmented images.** Ori means original image and Aug1, Aug2 and Aug3 are augmentated images. More detailed results are in the appendix.

```
25          return pos_enc
26
27      def forward(self, x, t):
28          t = t.unsqueeze(-1)
29          t = self.pos_encoding(t, self.time_dim)
30          return self.unet_forwad(x, t)
```

Our training strategy is as follows: E-step: 200 epochs → M-step: 400 epoch → E-step: 800 epoch. Continued training will further improve performance, but we did not increase the amount of computation due to computational resource constraints. The time loss of the method does improve due to the use of the diffusion model. However, on small datasets, this boost is acceptable. In this way at the same time DiffAug gives the possibility to accomplish unsupervised comparison learning training on small datasets.

Table A.3: Details of the training process in vision dataset.

| CF10 | $\nu$ | Learning Rate | Weight Decay | Batch Size | GPU | Training Time |
|------|-------|---------------|--------------|------------|-----|---------------|
| CF10 | 1 | 0.001 | 1e-6 | 256 | 1*V100 | 7.1 hours |
| CF100 | 2 | 0.001 | 1e-6 | 256 | 1*V100 | 7.2 hours |
| STL10 | 5 | 0.001 | 1e-6 | 256 | 1*V100 | 15.1 hours |
| TINet | 3 | 0.001 | 1e-6 | 256 | 1*V100 | 20.6 hours |

### C.3 DATA AUGMENTATION OF THE COMPARED METHODS

**BYOL augmentation.** The BYOL augmentation method is a hand-designed method. It is composed of four parts: random cropping, left-right flip, color ji

- Random cropping: A random patch of the image is selected, with an area uniformly sampled between 8% and 100% of that of the original image, and an aspect ratio logarithmically sampled between 3/4 and 4/3. This patch is then resized to the target size of 224 × 224 using bicubic interpolation.
- Optional left-right flip.

- Color jittering: The brightness, contrast, saturation, and hue of the image are shifted by a uniformly random offset applied to all the pixels of the same image. The order in which these shifts are performed is randomly selected for each patch.

- Color dropping: An optional conversion to grayscale. When applied, the output intensity for a pixel (r, g, b) corresponds to its luma component, computed as 0.2989r + 0.5870g + 0.1140b1.

**SimCLR augmentation.**

- Random Cropping: This involves taking a random crop of the image and then resizing it back to the original size. This can be seen as a combination of zooming and spatial location changes.

- Random Flipping: Randomly flip the image horizontally.

- Color Distortion: Apply a random color distortion. In the SimCLR paper, they use a combination of random brightness, random contrast, random saturation, and random hue changes. The strength of these distortions is controlled by a factor.

- Gaussian Blur: Apply a random Gaussian blur to the image. The extent of blurring is controlled by a factor.

**MoCo v2 augmentation.** For MoCo v2, the data augmentations are similar to those used in SimCLR, but there might be slight differences in implementation details. Here are the main augmentations used in MoCo v2:

- Random Cropping: This involves taking a random crop of the image and then resizing it back to the original size. Random Flipping: Randomly flip the image horizontally.

- Color Jitter: Randomly change the brightness, contrast, saturation, and hue of the image.

- Gaussian Blur: Apply Gaussian blur to the image with a certain probability.

- Solarization: This is an augmentation introduced in MoCo v2. It inverts pixel values above a threshold, which can create a unique visual effect.

**MAE augmentation.** The core idea behind MAE is to mask out parts of an image and then train an autoencoder to reconstruct the original image from the masked version. This is somewhat analogous to the masked language modeling task used in models like BERT for NLP, where parts of the text are masked out and the model is trained to predict the masked words.

## D  APPENDIX: DETAILS OF BIOLOGY EXPERIMENTS

### D.1  DATASET SETUPS

Experiments are performed on biological datasets, including MC1374[5] (Han et al., 2018), GA1457[6] (Rouillard et al., 2016), SAM[7] (Weber & Robinson, 2016), , and HCL500[8] (Han et al., 2020) datasets.

To ensure a fair comparison, we first embed the data into a 2D space using the method under evaluation. We then assess the method's performance through 10-fold cross-validation. Classification accuracy is determined by applying a linear SVM classifier in the latent space, while clustering accuracy is gauged using k-means clustering in the same space. Further details about the datasets, baseline methods, and evaluation metrics can be found in Table A.4.

---

[5]https://bis.zju.edu.cn/MCA/

[6]https://maayanlab.cloud/Harmonizome/gene/GAST

[7]https://github.com/abbioinfo/CyAnno

[8]https://db.cngb.org/HCL/

Table A.4: Datasets information of simple manifold embedding task

| Dataset | Train Samples | Test Samples | Input Dimension | Class Number in label |
|---------|---------------|--------------|-----------------|-----------------------|
| MC1374  | 24,000        | 6,000        | 1,374           | 98                    |
| GA1457  | 8,510         | 2,127        | 1,457           | 49                    |
| SAM561  | 69,491        | 17,373       | 561             | 52                    |
| HCL500  | 48,000        | 12,000       | 500             | 45                    |

## D.2 Baseline Methods and Implementation Details

Dimension reduction methods that have been widely used on biological analyze are compared, including kPCA (Halko et al., 2010), Ivis (Szubert et al., 2019), PHATE (Moon & van Dijk, 2019), PUMAP (Sainburg et al., 2021), PaCMAP (Wang et al., 2022), DMTEV (Zang et al., 2022a) and hNNE (Sarfraz et al., 2022).

For DiffAug, both the semantic encoder $Enc(\cdot)$, and the diffusion generator $Gen(\cdot)$, are implemented using a Multi-Layer Perceptron (MLP). Their respective architectures are defined as: $Enc(\cdot)$: [-1, 500, 300, 80]. The $Gen(\cdot)$: is defined below,

Listing 2: DiffusionModel for Biology Task

```
1    class AE(nn.Module):
2    def __init__( self,in_dim, mid_dim=2000, time_step=1000,):
3        super().__init__()
4        self.enc1 = self.diff_block(in_dim, mid_dim)
5        self.enc2 = self.diff_block(in_dim, mid_dim)
6        self.enc3 = self.diff_block(in_dim, mid_dim)
7        self.enc4 = self.diff_block(in_dim, mid_dim)
8
9        self.dec1 = self.diff_block(in_dim, mid_dim)
10       self.dec2 = self.diff_block(in_dim, mid_dim)
11       self.dec3 = self.diff_block(in_dim, mid_dim)
12       self.dec4 = self.diff_block(in_dim, mid_dim)
13       self.time_encode = nn.Embedding(time_step, in_dim)
14
15   def diff_block(in_dim, mid_dim):
16       return nn.Sequential(
17       nn.LeakyReLU(), nn.InstanceNorm1d(in_dim),
18       nn.Linear(in_dim, mid_dim), nn.LeakyReLU(),
19       nn.InstanceNorm1d(mid_dim), nn.Linear(mid_dim, in_dim),)
20
21   def forward(self, input, time, cond=None):
22       input_shape = input.shape
23       if len(input.size()) > 2:
24           input = input.view(input.size(0), -1)
25       ti = self.time_encode(time)
26       cd = self.cond_model(cond).reshape(input.shape[0], -1)
27       ee1 = self.enc1(input + ti + cd)
28       ee2 = self.enc2(ee1 + ti+ cd) + ee1
29       ee3 = self.enc3(ee2 + ti+ cd) + ee1 + ee2
30       ee4 = self.enc4(ee3 + ti+ cd) + ee1 + ee2 + ee3
31
32       ed1 = self.dec1(ee4 + ti+ cd)
33       ed2 = self.dec2(ed1 + ti+ cd) + ee3 + ed1
34       ed3 = self.dec3(ed2 + ti+ cd) + ee2 + ed1 + ed2
35       ed4 = self.dec4(ed3 + ti+ cd) + ee1 + ed1 + ed2 + ed3
36       return ed4.reshape(input_shape)
```

To assess the efficacy of the proposed methods, following Wang et al. (2022); Sarfraz et al. (2022), we utilized linear SVM performance to evaluate the performance of differences methods. For the linear SVM evaluation, embeddings were partitioned with 90% designated for training and 10% for

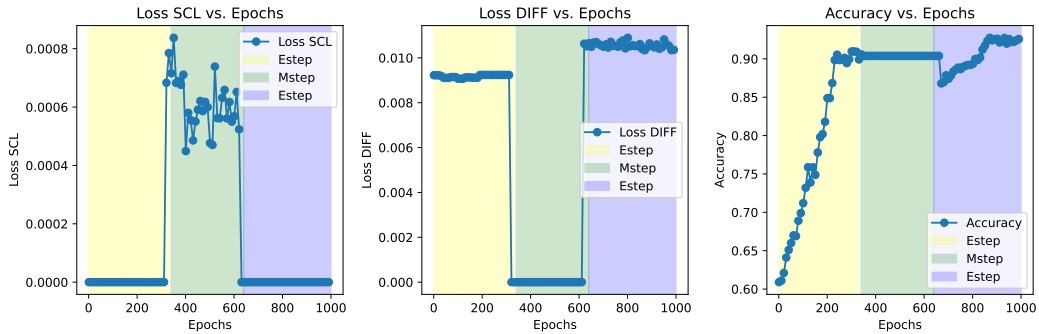

Figure A.2: Training curves on the GA1457 dataset, including two Esteps and one Mstep. We can observe that the new generated data improves the correctness of E step.

testing; the training set facilitated the linear SVM training, while the test set yielded the performance metrics. Detailed specifics of this configuration are elaborated in the Table A.5.

Table A.5: Details of the training process in biological dataset.

| CF10 | $\nu$ | Learning Rate | Weight Decay | Batch Size | GPU | Training Time |
|---|---|---|---|---|---|---|
| MC1374 | 1 | 0.0001 | 1e-6 | 300 | 1*V100 | 4.2 hours |
| GA1457 | 1 | 0.0001 | 1e-6 | 300 | 1*V100 | 4.6 hours |
| SAM561 | 1 | 0.0001 | 1e-6 | 300 | 1*V100 | 12.1 hours |
| HCL500 | 0.1 | 0.0001 | 1e-6 | 300 | 1*V100 | 20.1 hours |

