# OpenReview forum: "Boosting Unsupervised Contrastive Learning Using Diffusion-Based Data Augmentation From Scratch"
_ICLR.cc/2024/Conference — ICLR 2024 Conference Withdrawn Submission_

### Official Review · Reviewer_RifC · 2023-10-30

**Soundness:** 2 fair
**Presentation:** 3 good
**Contribution:** 2 fair
**Rating:** 3
**Confidence:** 4

**Summary:**

The paper introduces DiffAug, as an unsupervised diffusion-based data generator for producing data augmentation in contrastive learning framework. It aims to study how to automatically generate data augmentation that could replace traditional domain-specific and hand-crafted data augmentation. The proposed method is applied on both vision datasets and biological datasets, demonstrating improvements over previous SOTA.

**Strengths:**

- The paper aims to tackle a challenging problem, which was investigated in many previous works.
- The presented work demonstrated significant gain in biological datasets, demonstrating its usefulness in biomedical domains.

**Weaknesses:**

- **Lack of novelty and insights** The paper aims to address unsupervised data augmentation generation problem, which is widely studied and investigated. It remains unclear to me how the proposed approach can avoid mode collapsing of the generated data, as well as what kind of additional augmentation can be generated through the diffusion generator. In other words, why a diffusion generator outperform other generators? Is there any underlying reason that enforces the diffusion generator to produce good data augmentations?

- **Insufficient benchmarking**
1. The paper lacks computational costs analysis. How many parameters does the introduced diffusion generator contain? Would other baselines outperform the introduced method given the additional parameters?
2. The presented baseline performance on TinyImagenet is too low, see [1] which open-source 51%-acc BYOL baseline (and 92%-acc BYOL baseline on STL10). The performance gain demonstrated over non-SOTA baselines undermines its contribution.
3. The authors did not include the specs of the encoder backbone. I tried searching for ResNet over the text, and the only information I found is “We used the ResNet-50 (He et al., 2015) backbone” from the Appendix. However, In Simsiam paper they reported 91.1%-acc w SimCLR; and 91.8%-acc w SimSiam using a ResNet-18 on CIFAR10. Are the numbers In Table 1 produced from ResNet-18 or ResNet-50? Are the numbers from DiffAug produced by ResNet18 or ResNet50?
4. Does the method scale? Could the proposed method demonstrate such performance gain on ImageNet?

[1] https://arxiv.org/pdf/2007.06346.pdf

**Questions:**

See above

---

### Official Review · Reviewer_VepM · 2023-10-31

**Soundness:** 3 good
**Presentation:** 2 fair
**Contribution:** 2 fair
**Rating:** 5
**Confidence:** 4

**Summary:**

This paper proposes a new method called DiffAug to generate positive samples in contrastive learning (CL), which aims to improve the performance of CL without extra data or labels and domain knowledge. More specifically, this method will pretrain a diffusion-based generator with the same dataset for CL in a self-supervised way. And the trained generator will be applied to generate the positive pair when training the backbone model for CL. Experimental results show that by incorporating the proposed new method into CL, the backbone model can outperform the baselines on visual and biological datasets.

**Strengths:**

The strengths are listed as follows.
1. The paper is well-motivated. The disadvantages of the existing data augmentations for CL are clearly shown in Figure 1, which are overcome by DiffAug.
2. The evaluation of DiffAug is made on both visual and biological datasets, which demonstrates its effectiveness in different domains.

**Weaknesses:**

The weaknesses are listed as follows.
1. The proposed method requires training a diffusion-based model on the given dataset from scratch, which is very costly and expensive.
2. The summary of the existing model-based methods of data augmentation is not accurate. There already exist works that utilize generative models to generate data for CL without external data or labels. Some of this type of works are listed as follows.
[1]. Jahanian, Ali, et al. *"Generative Models as a Data Source for Multiview Representation Learning."* International Conference on Learning Representations. 2021.
[2]. Wu, Yawen, et al. *"Synthetic data can also teach: Synthesizing effective data for unsupervised visual representation learning."* Proceedings of the AAAI Conference on Artificial Intelligence.
3. Some important technical details of the DiffAug are not clear. For example, Algorithm 1 does not clearly show the workflow to pretrain the encoder and decoder. The transition of state *S* is not shown and thus it is unknown when to execute the M-step and when to execute the E-step. Moreover, in the E-step, the training starts with traditional data augmentation at the beginning and then switches to using the data from the generator with a probability $\lambda$. These important details are also not shown in the pseudo-code. For more missing or unclear technical details, please refer to the section of questions.
4. In the section on results, DiffAug is not compared to the existing model-based methods without external data or labels.

**Questions:**

The questions are listed as follows.
1. What is the training cost to pretrain the encoder and decoder from scratch?
2. What is the extra computing overhead to generate the positive samples in CL?
3. When utilizing the trained generator for CL, is the only thing to use the generate the positive samples with the generator? Is there any other steps or tricks?
4. Is the encoder in Figure 2 also the backbone model to be trained in CL?
5. Does the performance of the proposed method rely on the diffusion model? if we use other generative models like GAN to serve as the generator, will the proposed method outperform the naive CL method?
6. In Table 1, what is the difference between the row of SimC.(MoCo.) + Diff and the row of DiffAug?

---

### Official Review · Reviewer_AdtB · 2023-11-01

**Soundness:** 3 good
**Presentation:** 2 fair
**Contribution:** 2 fair
**Rating:** 3
**Confidence:** 4

**Summary:**

The paper proposes a new diffusion based contrastive unsupervised learning algorithm. The idea is conventional contrastive learning algorithm is heavily relying on manually designed data augmentation. However, it might not be easy and straightforward to design a “good” data augmentation especially for non-vision data such as biological data. The proposed contrastive learning pipeline contains 2 major components, a feature extractor and a diffusion model to generate augmented data. The feature extractor (encoder) is trained with a soft contrastive loss while the diffusion model is trained using a diffusion loss.

**Strengths:**

The idea of using a diffusion model to generate augmented training samples is very interesting. It’s also quite inspiring to see an iterative training strategy and the soft contrastive loss both boost the performance.

The paper is mainly clear and easy to follow. Although the benchmark datasets are small, the performance boosts for both vision and biological datasets are significant.

**Weaknesses:**

1. One weakness of the paper is the selection of the baseline methods. Most of the selected baselines are relatively old, the paper doesn’t compare with many of the newer self-supervised learning methods like DINO, MAE and some of the ViT self-supervised learning methods like BEIT (https://arxiv.org/abs/2106.08254) and EVA (https://arxiv.org/abs/2211.07636).

2. Another weakness is the selection of the benchmark datasets. Generally, the self-supervised learning method is applied to a relatively large dataset which is hard to collect supervised label like ImageNet or LAION. Due to computation constraints, the paper only compares the baseline methods on small datasets like CIFAR and Tiny-ImageNet.

3. A huge limitation of the proposed method is the computation cost. For conventional data augmentation methods like flipping or cropping, the computation cost is negligible compared to the model inference time. However, for the proposed method, in order to get one augmented method, it needs to pass through the whole diffusion process, which is quite expensive. It’s one of the reasons why the proposed method can only apply to small-scale datasets. In my opinion, the computation limitation should be clearly mentioned in the main paper instead of in the appendix. It might be helpful to save the computation by  pre-computing several (say 10) augmented samples for each training sample after training the diffusion model, instead of running the diffusion model on-the-fly.

4. The training stability is generally a pain-point for iterative optimization. I found very little discussion in the paper for the training stability. Sec. 3.1 and Sec. 3.2 mentioned different training strategies for vision data and biological data. It’s not clear whether the final performance is sensitive to the training strategy or not. Also, one minor thing, I don’t think it’s proper to name the proposed iterative training strategy EM algorithm (which is used to maximize the likelihood).

5. In the contribution part, the paper claims that “DiffAug’s data augmentation replaces traditional domain-specific hand-designed data augmentation strategy”. I don’t think it’s true, according to Section 2.2, DiffAug can only replace \lambda of the augmented data, which means the traditional data augmentation strategy is still needed. Section 3.5 discussed the selection of \lambda, which seems very important to understand this. However, Fig 7 in Section 3.5 seems incorrect.

A few minor things:

In Figure 6, it’s not clear which embedding is better. It’s too small and hard to read.

In Section 3.3, the paper mentioned “DiffAug’s similarity distribution is smoother and broader.” It’s not clear to me 1) what’s the meaning of smoother and broader for a distribution and 2) why smoother and broader are better.

**Questions:**

My major concerns of the papers are the computation cost, see weakness 3 and training stability see weakness 4. Having a discussion on these 2 points will help me better understand the context and the novelty of the paper.

---

### Official Review · Reviewer_Dx6S · 2023-11-01

**Soundness:** 3 good
**Presentation:** 3 good
**Contribution:** 2 fair
**Rating:** 5
**Confidence:** 5

**Summary:**

This paper introduces a novel technique called DiffAug, which is designed to improve representation learning in an unsupervised and uninterrupted manner. DiffAug achieves this by iteratively training a semantic encoder and diffusion model, which outperforms hand-designed augmentation in classical contrastive learning methods. The paper also includes experimental evaluations on both image and biological data that demonstrate the effectiveness of DiffAug.

**Strengths:**

- This paper is clearly written and the idea is easy to follow.
- The visual illustration is clear and informative, which helps the readers understand the main idea.
- The experimental results show the superiority of this method, where DiffAug outperforms all the baselines on image and biology data.
- The experiments across different modalities and domains show the generalizability and effectiveness of this method.

**Weaknesses:**

- Lack of some related works: (1) the idea of using diffusion models to enhance the augmentation process of contrastive learning has been studied in the graph domain in a previous work [1], which however has not been properly cited; (2) the pipeline of training diffusion models with a latent representation has been widely discussed in many previous works [2, 3, 4] and they need to be properly cited.

- The details of the used biological datasets should be better discussed either in the main text or the appendix, e.g., what the resources and modalities of the data are, how we construct the model input from those data, how these data look like, and what the meaning of those data is, etc.
- Diffusion models are slow during training and sampling compared to simple augmentation baselines. The running time analysis and comparison should be included in the main text to let people understand more about the trade-offs between performance gain and time costs. The data used in this paper seems to be low quality (resolution) which somehow keeps relatively low training costs. I would like to see how the computational costs scale up when the high-quality data comes.

- Some typos/unclear statements: (1) in the paragraph above equation (5), "data generated by DiffAug is replaced by ...";
(2) above equation (4), DDMP -> DDPM;
(3)  in equation (3), A_{ug} -> Aug; ...

[1] Liu, Gang, et al. "Data-Centric Learning from Unlabeled Graphs with Diffusion Model." arXiv preprint arXiv:2303.10108 (2023).

[2] Preechakul, Konpat, et al. "Diffusion autoencoders: Toward a meaningful and decodable representation." Proceedings of the IEEE/CVF Conference on Computer Vision and Pattern Recognition. 2022.

[3] Wang, Yingheng, et al. "InfoDiffusion: Representation Learning Using Information Maximizing Diffusion Models." arXiv preprint arXiv:2306.08757 (2023).

[4] Zhang, Zijian, Zhou Zhao, and Zhijie Lin. "Unsupervised representation learning from pre-trained diffusion probabilistic models." Advances in Neural Information Processing Systems 35 (2022): 22117-22130.

**Questions:**

- Why do we need EM training? Can we do finetuning on a pre-trained diffusion model? Can we first train a good latent diffusion model and then use it to generate good augmentations for contrastive learning? Would some regularization terms (like prior regularizations) increasing the diversity of generated augmentations benefit contrastive learning? How does the design choice of latent diffusion architecture affect the performance? (To me, it seems the latent representation highly influences the generation quality of diffusion models. Thus, how we fuse it inside of the networks could be important to investigate.) Overall, the strategies need to be studied and discussed.

- As I mentioned above, the running time (both training and sampling) is essential to investigate. Since some hand-crafted augmentations have achieved relatively good performance, if this method is far slower than the other baselines, the benefits of introducing another expensive training component seem to be unnecessary. What does the computational complexity look like?

- What are these biological datasets? Why would we want to compare the method with these simple unsupervised dimensionality reduction approaches on these biology data? How about other types of biological modalities? What about the performance of some similar self-supervised learning techniques on these datasets instead of those simple baselines?